# Evaluation of Undergraduate Learning Experiences in Pediatric Dentistry during the COVID-19 Pandemic

**DOI:** 10.3390/ijerph20032059

**Published:** 2023-01-23

**Authors:** Tamara Pawlaczyk-Kamieńska, Justyna Opydo-Szymaczek, Natalia Torlińska-Walkowiak, Beata Buraczyńska-Andrzejewska, Kinga Andrysiak-Karmińska, Dorota Burchardt, Karolina Gerreth

**Affiliations:** 1Department of Risk Group Dentistry, Pediatric Dentistry, Poznan University of Medical Sciences, 60-812 Poznan, Poland; 2Department Pediatric Dentistry, Poznan University of Medical Sciences, 60-812 Poznan, Poland; 3Center for Innovative Education Techniques, Poznan University of Medical Sciences, 60-755 Poznan, Poland; 4Department of Medical Education, Poznan University of Medical Sciences, 60-806 Poznan, Poland

**Keywords:** dental education, COVID-19, pediatric dentistry, learning outcomes

## Abstract

The aim of the study was to analyze students’ opinions on the learning outcomes they achieved during clinical classes in pediatric dentistry. The classes were run in various periods: before the SARS-CoV-2 coronavirus pandemic (onsite classes), in the first months of the pandemic (online classes), and in the following months of the pandemic (onsite classes with additional infection control and administrative changes in patients’ admission procedures). Material and methods. The research was conducted among fifth-year dentistry students at Poznan University of Medical Sciences. Students who completed the entire course and passed the diploma exam in pediatric dentistry were asked to complete the anonymous questionnaire providing their opinion. Results. The research results showed that, in the students’ opinion, clinical classes—regardless of their form—allowed them to achieve the knowledge necessary to perform pediatric dental procedures. However, the students appreciated onsite classes significantly more compared to information and communication technologies (ICT) classes in gaining practical skills and social competence. Conclusions. (1) The significant limitations introduced because of the SARS-CoV-2 pandemic impacted the development and implementation of modern online teaching techniques, which may very well be applied to convey theoretical knowledge after the pandemic has ended. (2) Skills and social competence, indispensable in the future dental practice of dental students, can only be obtained during onsite classes. (3) Medical universities should design standards of teaching to enable them to address a possible epidemiological threat in the future, which will enable rapid response and safe continuation of practical dental education during a pandemic.

## 1. Introduction

Dental education is a specific combination of various teaching methods that determine the organization of studies. They include presentation methods, such as lectures; problem methods, such as seminars; and practical methods, such as demonstrations and clinical classes [1,2]. It is important to emphasize that dental education is fundamentally different from medical education. Currently, the primary focus is on developing the practical skills of future dental practitioners. Students, strictly supervised by an experienced dentist, actively participate in clinical procedures, which enables them to gain knowledge and to master social competence (e.g., establishing and maintaining rapport with the patient, observing medical confidentiality, upholding patients’ rights and adhering to the principles of medical ethics, showing awareness of limitations resulting from a disease and the resulting social issues) and skills (e.g., diagnosing the most common diseases, assessing and describing the patient’s somatic and mental state, and conducting professional dental care, consisting of prophylaxis, treatment, health promotion, and health education) [1,3,4]. Therefore, a majority of courses take the form of clinical classes, with the students actually present, which cannot be replaced by information and communication technologies (ICT) format education.

On 11 March 2020, the World Health Organization (WHO) announced the existence of the coronavirus disease (COVID-19) pandemic [5], with a rapid rise in the number of cases and the spread of the SARS-CoV-2 coronavirus, an acute infectious disease of the respiratory system [5]. COVID-19 is a global health threat, which spread through respiratory droplets and aerosol transmission. The causative agent of this disease is the severe acute respiratory syndrome coronavirus-2 (SARS-CoV-2), an enveloped positive single-stranded RNA virus, responsible for infection in pulmonary and extrapulmonary organs [6]. To protect students, staff, and patients from infection, many medical universities worldwide had to decide how to continue their education and determine what methods of clinical teaching to implement. Moreover, in Poland, as in many other European countries, it was recommended to significantly limit the range of dental services provided, usually to necessary procedures, including those requiring urgent intervention, such as pain, inflammations and purulent conditions, cysts, and conditions likely to promote a disease progression and to produce complications [7].

The first months of the pandemic appeared difficult, with little knowledge about and much misinformation regarding the SARS-CoV-2 virus. Numerous medical universities in Europe, which—apart from education—are obliged to ensure the safety of their staff and students, decided to switch to distance learning entirely [8]. From this perspective, dental schools were particularly affected by the COVID-19 pandemic, due to the high risk of exposure of dental operators and dental students, during training practice. For these reasons, lectures and practical training were suspended and new methods of teaching were introduced [9]. Universities were hastily introducing modern, but untested in dental education, ICT technologies and they provided academics with digital platforms for remote teaching. Unfortunately, the implementation of those platforms was not preceded by any training, so both teachers and students gradually learned the new tools and how to use them. One advantage of these new techniques is that they helped maintain student–teacher contact during the pandemic lockdown.

Dentistry students should achieve specified learning outcomes in a given subject related to knowledge, skills, and social competence [1,4]. Knowledge is understood as the effect of acquiring information through learning (the student knows and understands a given issue). A skill is an ability to apply knowledge and use “know-how” to handle problem-solving tasks (that the student is able to do). Competence is the ability to apply knowledge, manual skills, and personal and social skills in a dental profession (the student is prepared to talk to a young patient and their caregivers) [1]. Therefore, learning outcomes define what the learner should know, understand, and be able to do after they graduate and enter the profession. The goal is also to improve the manual dexterity of future dentists. It is impossible to fulfill this task with ICT education, and online classes are not a substitute for real patient–student contact. Therefore, after the summer break in 2020, which allowed universities to adapt to the new pandemic conditions, during the 2020/2021 academic year, many medical universities decided to conduct onsite clinical classes, including those in dentistry. However, resuming these classes posed a significant challenge. It was necessary to modify and adapt the didactic infrastructure to the sanitary-epidemic requirements related to biosecurity issues and to introduce a system to monitor the spread of infections and implement a social distance policy, which undoubtedly impacted the ability to conduct the clinical classes.

The study aimed to analyze the opinions of students regarding the learning outcomes (knowledge, skills, social competencies) they achieved during clinical classes in pediatric dentistry conducted in three different periods: before the pandemic, i.e., in the winter semester of the 2019/2020 academic year (onsite classes); in the first months of the pandemic, i.e., in the summer semester of the 2019/2020 academic year (online classes); and in the months following the COVID-19 pandemic, i.e., in the winter semester of the 2020/2021 academic year (onsite classes with added infection control and administrative changes in patient admission procedures, hereinafter referred to as onsite classes under the sanitary regime).

## 2. Material and Methods

The study was conducted among fifth-year dental students at Poznan University of Medical Sciences. The inclusion criteria for the study were: (1) completion of clinical classes in the three periods described above (onsite classes during semester VII, online classes during semester VIII, and onsite classes in sanitary regimen during semester IX), (2) earning credit for the entire course in pediatric dentistry, and (3) passing the diploma exam. The criteria for exclusion from the study were: (1) absence during any part of the semesters (the above-described periods) due to health reasons or dean’s leave.

Students who completed the entire course and passed the diploma exam in pediatric dentistry were asked to complete an anonymous questionnaire providing their opinions on the clinical classes in pediatric dentistry offered at the three different periods. Apart from demographic information, such as age and gender, the questionnaire contained questions prepared based on the selected detailed educational outcomes included in the syllabuses of the Doctor of Dental Surgery Program Didactic Guide the Poznan University of Medical Sciences [4]. The questions concerned the students’ views on the learning outcomes they achieved, in each of the analyzed periods, in relation to the knowledge required to undertake procedures in pediatric dentistry, selected skills, and designated social competence in pediatric dentistry. The questionnaire employed a 5-point Likert scale [10], consisting of a ceteris paribus of five statements ranging from total rejection to total acceptance.

Prior to the study, confirmation was obtained from the Chairman of the Bioethics Committee of the Poznan University of Medical Sciences that the survey described above is not a medical experiment and does not require an opinion of the Bioethics Committee. As all questions used in this study were developed by researchers specifically for this project and the recent outbreak of COVID-19, the questionnaire was not validated or piloted before this study.

In the first stage of statistical analysis, the analysis evaluated the students’ opinions of the general learning outcomes they achieved related to their (1) knowledge (knowing and understanding), (2) skills (knowing how to act), and (3) social competence (values, knowing how to be). In the second stage, the respondents’ opinions as to the selected detailed educational results were analyzed.

To compare the students’ replies to the same questions that were asked after completing classes taught in different settings (onsite, online, and onsite under the sanitary regimen), Friedman’s ANOVA was applied. When the results of that analysis were statistically significant, specific differences were explored using the Conover–Inman post hoc test. The data were described in tables by giving the number and percentage of each category and in figures giving medians, quartiles, minimum and maximum values. The results were considered statistically significant when the significance level was *p* < 0.05. All statistical analyzes were performed in PQStat v1.8.4.

## 3. Results

### 3.1. Demographic Data

A total of 158 students who met the inclusion criteria were invited to participate in the study. The questionnaires were completed by 63.92% (101) of the students. The mean age of the students was 25.54 ± 2.93. Females constituted 66.34% of all subjects.

### 3.2. Evaluation of the Achieved Learning Outcomes in the Scope of Knowledge

The result evaluating the acquisition of knowledge necessary to undertake procedures in pediatric dentistry, during various forms of teaching, in the students’ opinion was inconclusive.

Statistical analysis showed that in terms of the knowledge acquired, students rated practical classes in the sanitary regime higher than online classes (*p* = 0.0039). In the students’ opinion, the face-to-face classes conducted before and during the pandemic enabled them to achieve comparable levels of knowledge learning outcomes (*p* > 0.05) (Figure 1).

Although Friedman’s test showed statistically significant differences between the three types of education (*p* = 0.0162), a post hoc test was not statistically significant (*p*-value > 0.05) (Figure 1). Among the respondents, 76.8% rated the onsite classes positively, 68.7% rated the online classes positively, and 87.9% rated the onsite classes under the sanitary regimen positively (Table 1).

### 3.3. Evaluation of Achieved Learning Outcomes in the Scope of Skills

Statistical analysis of the students’ opinions on their ability to achieve the learning outcomes during practical classes in all six studied skills showed that there were significant differences between different types of classes, i.e., onsite classes before the pandemic versus online classes (*p* = 0.0325), online classes versus onsite classes during the pandemic (*p* < 0.0001), and between the onsite classes before and during the pandemic (*p* = 0.0107) (Figure 2).

Moreover, 69.43% of the respondents rated the onsite classes positively, 79.05% rated the onsite classes under the sanitary regimen positively, and 55.86% rated the online classes positively; 11.36%, 7.9%, and 25.07% of the respondents rated those classes negatively, respectively (Table 1).

When analyzing the students’ opinions about their ability to achieve specific learning outcomes in the scope of skills, a statistically significant difference between the three examined periods was noted for the following skills: (1) clinical diagnostics of a patient of developmental age (Figure 3A), (2) prevention of oral diseases and conducting dental prophylaxis in a patient of developmental age (Figure 3C), and (3) treatment of dental diseases in patients of developmental age (Figure 3D). Regardless of the form of instruction, the students rated onsite classes significantly higher than online classes (*p* < 0.05). However, there was no difference between the onsite pre-pandemic and pandemic classes (*p* ≥ 0.05).

Statistically significant differences between the three study periods were also noted regarding the acquisition of skills, specifically skills in using appropriate methods and medications during and after the dental procedure in order to eliminate pain and anxiety in an adolescent patient (Figure 3B); planning comprehensive treatment of the diseases of teeth, periodontal tissues, and oral mucosa in children and adolescents (Figure 3E); and in keeping records of an adolescent patient, writing letters of referral for them for specialized dental care, conducting general medical examinations, and providing treatment (Figure 3F). A significant difference was only noted between the online and onsite classes under the sanitary regimen. In contrast, there was no significant difference between the onsite classes conducted before the pandemic outbreak and the online practice classes.

Statistical analysis showed that in terms of the acquired skills in planning comprehensive treatment of the diseases of teeth, periodontal tissues, and oral mucosa in children and adolescents, students rated practical classes in the sanitary regime higher than online classes (*p* = 0.0003). The face-to-face classes conducted before and during the pandemic enabled them to achieve comparable levels of these skills (*p* > 0.05) (Figure 3).

Statistical analysis showed statistically significant differences between the three study periods regarding the students’ opinions about acquiring skills in planning comprehensive treatment of the diseases of teeth, periodontal tissues, and oral mucosa in children and adolescents (Figure 3E) (*p* ≥ 0.05). The percentage of students that were positive about their ability to achieve this outcome during onsite, online, and onsite classes under the sanitary regimen was similar (58.1%, 51%, and 64.3%, respectively) (Table 1).

### 3.4. Evaluation of the Learning Outcomes Achieved in Social Competence

Statistical analysis of the results revealed that the students rated their chances of obtaining the required learning outcomes in the two studied social competencies to be higher during practical classes conducted onsite (before and during the pandemic) in comparison to ICT classes (*p* < 0.0001) (Figure 4).

Moreover, in the students’ perceptions, the face-to-face classes conducted before the pandemic and the face-to-face classes during the pandemic enabled them to master these competencies to a comparable extent (*p* > 0.05). Furthermore, 66.35% of the respondents rated the opportunity to acquire social competencies during onsite classes held before the outbreak of the COVID-19 pandemic as high; similarly, 77.4% of the respondents rated the opportunity to acquire social competencies in onsite classes held during the pandemic as high, while only 45.25% rated the opportunity to acquire social competencies during online classes as high. A negative rating was given to onsite classes before the pandemic and during the pandemic and to online classes by 6.5%, 5%, and 23.1% of the respondents, respectively (Table 1).

Similar results were obtained by analyzing the students’ opinions regarding specific social competencies, such as communicating with a patient of developmental age and their parents (Figure 5A) and the competence of taking medical history from a patient and/or their family (Figure 5B). The students rated the opportunity to acquire these competencies similarly in onsite classes before the pandemic and in onsite classes during the pandemic (*p* ≥ 0.05), but the students assigned a significantly lower rating to the opportunity to acquire these competencies in online classes (*p* < 0.05).

### 3.5. Evaluation of the Achieved Learning Outcomes

Out of all evaluated learning outcomes, the students assigned the highest rating to the ability to prevent oral diseases and conduct dental prophylaxis in a patient of developmental age. Of the respondents, 83.3% rated the onsite classes positively, 71.8% rated the online classes positively, and 92.7% rated the onsite classes under the sanitary regimen positively (Table 1). These were the highest ratings given by the students in each of the study periods. However, they gave the lowest ratings to the acquisition of skills in keeping records for patients of developmental age, writing referrals for them for specialist dental care, conducting general medical examinations, and providing treatment. Negative ratings were given by 18.6% of the students for onsite classes, 32% of the students for online classes, and 15.5% of the students for onsite classes under the sanitary regimen, and these were also the lowest ratings in each period (Table 1).

## 4. Discussion

The aim of teaching dentistry is to equip graduates with the knowledge, skills, and professional competence in accordance with the standards of the European Union [1,2]. Therefore, while clinical classes must meet educational standards that prepare students to practice dentistry, they must also consider how to adapt to the pandemic recommendations that were in force at the time.

In their studies, students should achieve learning outcomes regarding knowledge, skills, and social competence determined by the curricula, which will prepare them for independent work [1]. In the first half of the 2019/2020 academic year, classes at Poznan University of Medical Sciences were held in compliance with the current study programs. Dental students actively participate in preventive and therapeutic procedures. Unfortunately for the fourth-year students, only half of the practical courses in pediatric dentistry were taught in this format. Due to the outbreak of the COVID-19 pandemic, the second part of the practical coursework was conducted using an ICT format. Classes included presentations and discussions of clinical cases prepared by doctors or students, joint discussion of the presented clinical problem and drawing up a treatment plan, remote participation of students in procedures performed in real time by the teacher, educational films, and independent student work. The problem of continuing dental education in the first months of the COVID-19 pandemic in a conventional way was shared by all European countries, as also shown by the survey conducted by Quinn et al. (2020) [8]. In all 69 dental schools across Europe that participated in the survey, restrictions were implemented to limit the spread of the SARS-CoV-2 virus; 95% of the dental schools implemented online teaching, 72% used live or streamed videos, 48% used materials available online, and 65% held virtual meetings with students [8].

Summer break allowed dental schools to adapt their teaching to the prevailing pandemic conditions. Due to the nature of the care provided, a significant challenge for clinical dental education, usually provided in multi-station offices, is maintaining social distance. Classes bring together physicians, students, and patients in a relatively small space. When the patients are minors, their guardians are also included. Moreover, routine dental procedures involve the release of aerosols, which are generated primarily by handpieces, three-in-one syringes, and ultrasonic scalers. Water, combined with body fluids (blood, saliva), forms bioaerosols [11,12], which constitute a route of coronavirus transmission. Especially in the closed environment of a dental office, the viral concentrations are high [11] and persist for a long time [13,14]. It should also be noted that procedures performed with dental students participating take much longer than those performed by experienced dentists. Due to COVID-19 transmission, the universities needed to implement additional infection controls in clinical classes and administrative changes in patient admission procedures within a short period of time.

Above all, a significant challenge was to limit the spread of the bioaerosols emitted by the patient during medical services provision, especially since the patient presenting could be an asymptomatic carrier of COVID-19, which was the case for most children at the beginning of the pandemic. It was imperative that extra infection control measures be implemented and strictly observed by all [5,7]. All staff members (physicians, dental nurses), students, and patients had their body temperature taken with a contactless thermometer. Moreover, patients and their caregivers completed questionnaires on their general health status, particularly concerning respiratory symptoms. Patients who did not show symptoms and were not at risk of infection (those who had contact with a sick person) were treated as low-risk for COVID-19 according to WHO guidelines [15] and were admitted to the clinic. It was also necessary to institute a social distance policy, which is very challenging during clinical sessions with students. To limit the number of people in one place at the same time, a decision was made to control patient flow, limit the number of patients, and use accessory shielding between units to isolate individual patients from each other. Because of the high risk of coronavirus transmission during dental procedures [12], extra precautions were taken. All medical personnel (including students) were equipped with disposable long-sleeved gowns, surgical caps, shoe protectors, FFP2 masks, goggles or glasses, face shields, and disposable gloves [7]. Extra ventilation in the rooms was also mandatory. Apart from the standard air exchange, air purifiers were installed in the offices, switched on during a dental visit, and kept on for a period after the visit.

Implementing additional hygienic and epidemiological restrictions related to the COVID-19 pandemic allowed students to return to dental offices. Students could re-establish face-to-face contact with patients, perform physical examinations of patients, take a patient’s history, and actively participate in providing preventive and therapeutic procedures.

Our survey of the students who had the opportunity to take onsite clinical classes before the pandemic, online clinical classes after the pandemic outbreak, and onsite classes during the pandemic allowed us to determine their opinions about the learning outcomes they achieved in pediatric dentistry training before their diploma exam. This group of students encountered all three methods of instruction in the same subject. To date, the study presented here is the only one to report on data obtained from a survey of dental students about their opinions of the efficacy of online versus onsite dental education classes before, during, and after the COVID-19 pandemic. The available literature to date only addresses the evaluation of online classes during the COVID-19 pandemic [16,17,18,19,20], notable among which is the 2021 study by Herr et al. [18] comparing students’ opinions of onsite pediatric dentistry classes prior to the pandemic with their views on the same subject conducted in online classes during the pandemic. Unlike our study, which focused on clinical classes, the research cited [18] only included theoretical classes. The majority of the respondents in this study (92.3% of the 220 respondents) felt that the introduction of online classes during the first months of the pandemic was the right decision; 74.1% of the students were satisfied with the format of classes and believed that offline and online classes were comparable in many respects. Similar results regarding students’ evaluations of online theory classes during the pandemic were reported by Hassan et al. [16], Hattar et al. [17], and Jiang et al. [19]. In all these studies, the majority of students were satisfied with online lectures (73.5% of the 377 respondents in Hassan et al. [16], 86.5% of the 104 respondents in Jiang et al. [19], 67.1% of the 310 respondents in Hatter et al. [17], and 70% of respondents in Paolone at al. [20]) and would prefer this format in the future.

Moreover, according to Jiang et al. [19], lectures were the predominant form of online classes (approximately 39% of all online classes). Our own research findings confirm the reports of the studies cited above. Although we did not evaluate theoretical classes, the respondents assigned a high rating to the chance to gain knowledge that is essential for performing clinical procedures in pediatric dentistry during the ICT classes. Furthermore, a similar percentage of students positively evaluated the acquisition of knowledge during all the clinical classes, regardless of the format in which they were conducted. It seems that e-learning is as effective as traditional education held in a lecture hall and, in the opinion of the students, it allowed them to achieve the educational outcomes required by the curriculum. Moreover, as shown by the cited studies, students prefer this format for theoretical classes.

However, the students’ feedback on practical clinical classes conducted using ICT is different. The literature reports that despite various formats (e.g., presentations and discussions of clinical cases or problem-solving), the students were not pleased [16,17,18,19,21,22]. Negative ratings were given by 60% to 86% of the students, depending on the study [16,17,18,19,21,22], with approximately 80% of the respondents stating that the COVID-19 pandemic adversely affected their clinical skills [16,17]. Furthermore, Hassan et al. [16] noted that 34% of the respondents linked the pandemic to fewer patients being admitted throughout the teaching process. The vast majority (approximately 90%) of the respondents in these studies expressed a desire and need for supplementary clinical training to compensate for the lost clinical classes [16,22]. The results of our study confirm the findings reported in other studies. More than half of the surveyed students in our study did not have an opinion on or disagreed with the statement that online classes allowed them to learn how to interview patients and their caregivers and to strengthen their ability to take a patient’s medical history. Similar results were obtained by analyzing the students’ opinions on acquiring skills in diagnosing, preventing, and treating oral diseases in patients of developmental age.

Previously published observations highlight the reduced quality of clinical classes conducted online during the first months of the COVID-19 pandemic [16,17,18,19,20,21,22]. Although the use of new technologies made it possible to continue the education of dental students, despite the great effort made by teachers and students, as shown by the results of the surveys, in the opinions of the students, they were unable to achieve the expected educational results in terms of skills and social competence to the extent that was comparable to the classes conducted before the outbreak of the COVID-19 pandemic. It can be justified by the fact that the sudden outbreak of the pandemic did not allow universities to prepare for this unpredictable teaching situation.

An interesting finding of our study is that there was no significant difference in mastering social competence skills and some of the skills (diagnostics, prevention, treatment planning, and treatment itself) needed to practice dentistry between onsite classes before the outbreak of the COVID-19 pandemic and onsite classes with extra infection controls and administrative changes in patients’ admissions during the pandemic. Despite the introduction of severe social constraints and the need for social distancing and, consequently, fewer patients in clinical classes and the resulting smaller number of procedures conducted by the dental students, the students assigned equally high ratings to the chance to obtain the learning outcomes in competence and the previously mentioned skills both in onsite clinical classes and in onsite clinical sanitary regimen classes. The students appear to understand and accept the need for numerous sanitary-epidemiological restrictions in onsite clinical classes during the COVID-19 pandemic.

Interestingly, our studies noted a significant difference between onsite classes taught under a sanitary regimen and online classes in terms of the ability to acquire skills in using appropriate methods and medications during and after dental procedures to relieve an adolescent patient’s pain and anxiety, as well as skills in keeping records for an adolescent patient, writing a referral for the patient for specialized dental care, conducting general medical examinations, and providing treatment. However, there was no significant difference between onsite classes conducted before the pandemic outbreak and online practical classes. It may be explained by the fact that these skills require a more extended training period and more patient contact during which students have the opportunity to refine their skills.

The reported significant variations in acquiring the studied skills and social competence between onsite classes—irrespective of the prevailing epidemic conditions—and online classes indicate that it is essential to interact with the patient in a dental educational teaching environment. It will ensure not only the acquisition and implementation of the skills required to conduct medical procedures but also the ability to master effective communication with the patient, i.e., the ability to listen to and use terms understandable to the patient and to learn how to take the patient’s medical history.

The study results also indicate that universities should address the students’ need to acquire certain skills which—in their opinion—were not mastered well enough (regardless of the form of education), such as keeping records of patients of developmental age, writing referrals for examinations or specialized dental care, and providing general medical treatment. Such information is a valuable reminder when preparing classes for the following years.

It should be emphasized that the present research has some strengths since the survey was attended by students who completed the entire course and passed the diploma exam in pediatric dentistry. After the study, all respondents no longer had classes in pediatric dentistry, therefore, we may suspect that their answers were completely sincere. In addition, before the study, all students were informed that the data collection is anonymous and that it is impossible to identify the person providing the answer. In this way, bias in answering was prevented. A limitation of the study is that it was conducted at only one dental school. The second limitation is the size of the research sample. The questionnaires were completed by 64% of the students invited to participate in the study. In this study, students only expressed opinions on the learning outcomes they achieved during clinical classes in pediatric dentistry during the pandemic period. These were the subjective opinions of students.

## 5. Conclusions

The survey conducted for this study showed that the COVID-19 pandemic affected dental education. On the one hand, considerable restrictions were introduced that allowed for the development of modern remote learning techniques and the implementation of them into teaching, which can be used for theoretical classes, such as lectures or seminars, even after the pandemic has ended. On the other hand, studies have shown that there can be no substitute for one-to-one contact with the patient and, as in the case of pediatric dentistry, their caregivers. Live communication allows dental students to acquire skills and social competence in establishing rapport with their patients, taking medical histories, maintaining medical records, and providing prophylactic and therapeutic services. All of the educational outcomes discussed in this paper are vital to the students’ future dental profession. Therefore, medical universities should design standards of teaching to enable them to address a possible epidemiological threat in the future, which will enable rapid response and safe continuation of practical dental education during a pandemic.

## Figures and Tables

**Figure 1 ijerph-20-02059-f001:**
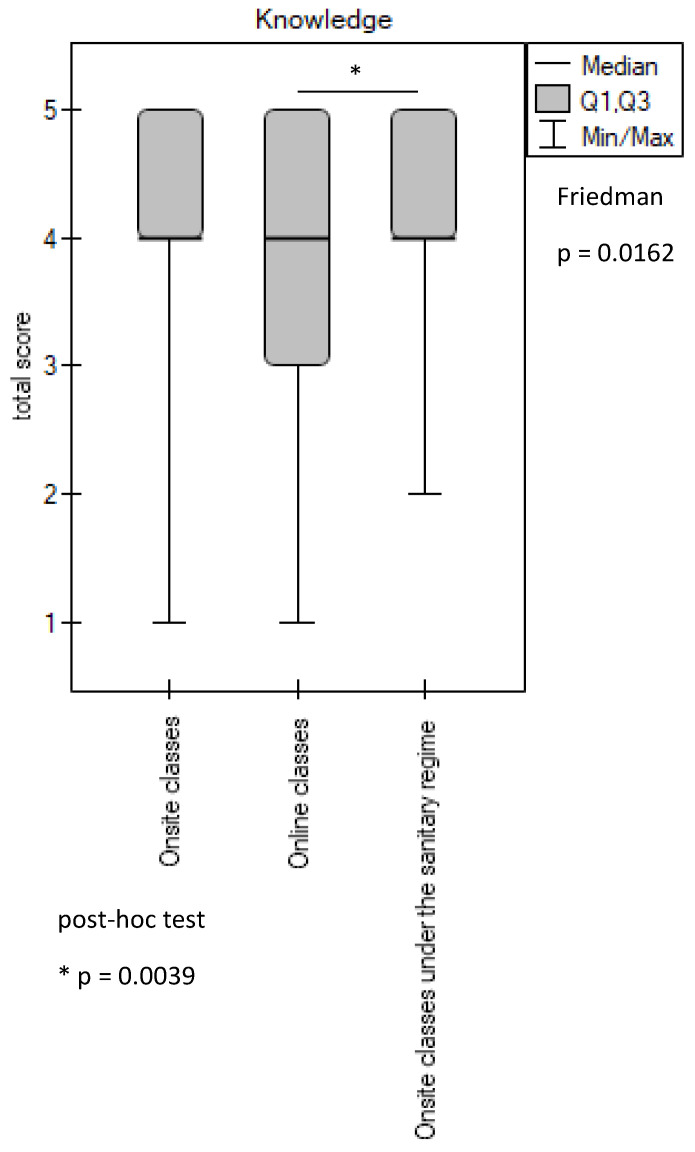
Evaluation of the achieved learning outcomes in the scope of knowledge. * *p*-value is given for statistically significantly different groups. For groups with no significant differences (*p* ≥ 0.05), the *p*-values are not quoted.

**Figure 2 ijerph-20-02059-f002:**
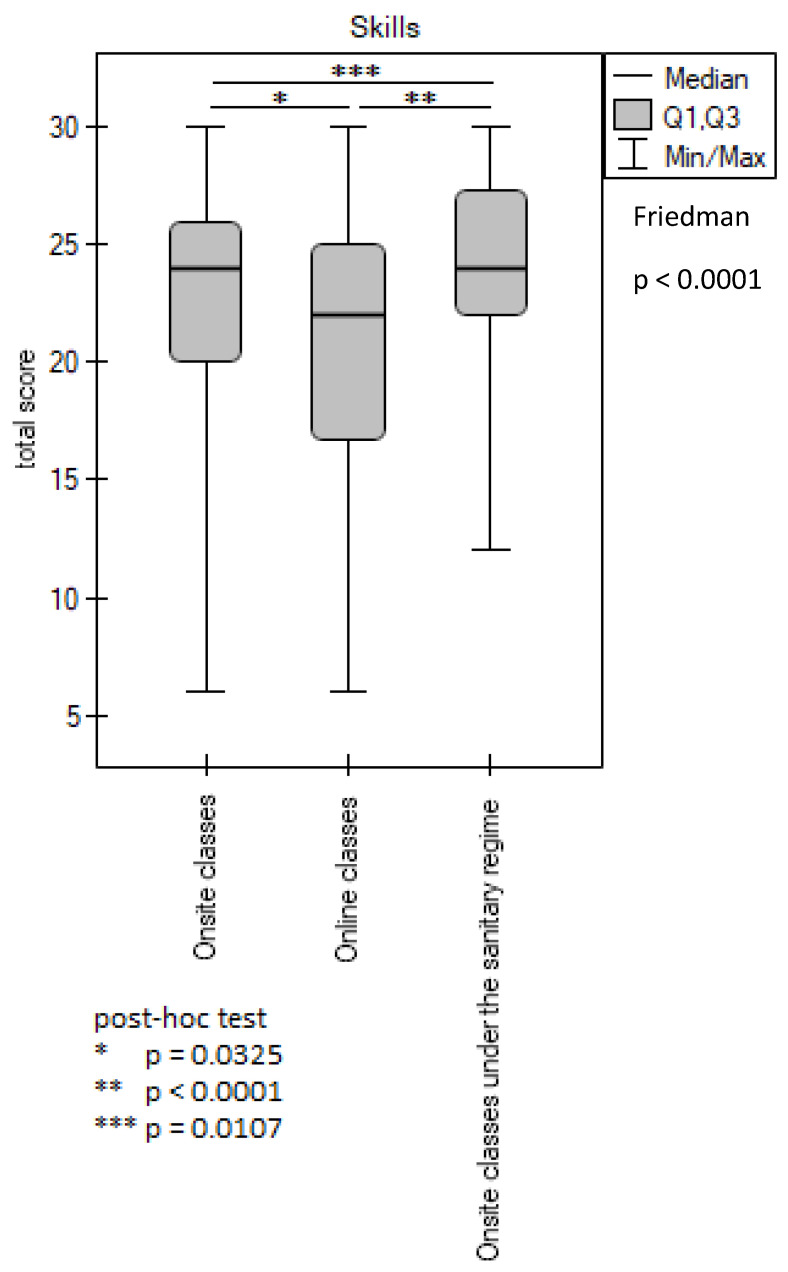
Evaluation of the achieved learning outcomes in the scope of skills. *, **, *** *p*-value is given for statistically significantly different groups.

**Figure 3 ijerph-20-02059-f003:**
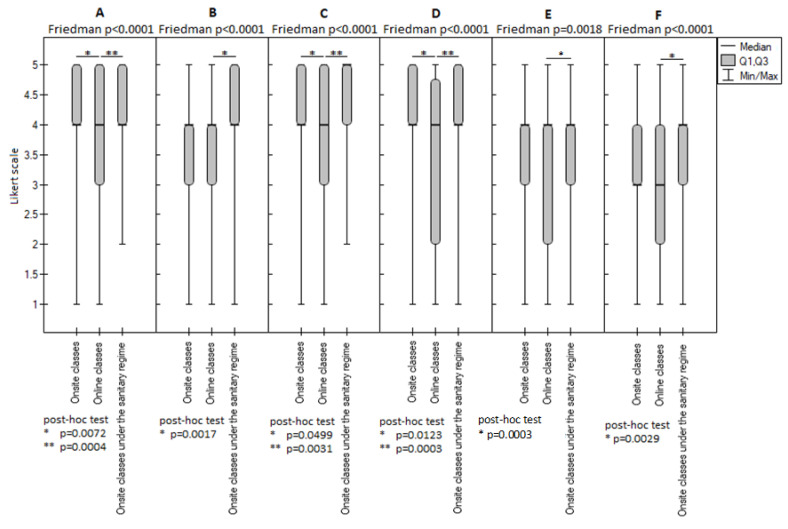
Evaluation of the achieved learning outcomes in the scope of: (**A**) clinical diagnostics of patients of developmental age; (**B**) applying appropriate methods and medications during and following dental procedures to relieve pain and anxiety in an adolescent patient; (**C**) prevention of oral diseases and conducting dental prophylaxis in patients of developmental age; (**D**) treatment of dental diseases in patients of developmental age; (**E**) planning comprehensive treatment of the diseases of teeth, periodontal tissues, and oral mucosa in children and adolescents; (**F**) keeping records of an adolescent patient, writing letters of referral for them for specialized dental care, conducting general medical examinations, and providing treatment. *, ** *p*-value is given for statistically significantly different groups. For groups with no significant differences (*p* ≥ 0.05), the *p*-values are not quoted.

**Figure 4 ijerph-20-02059-f004:**
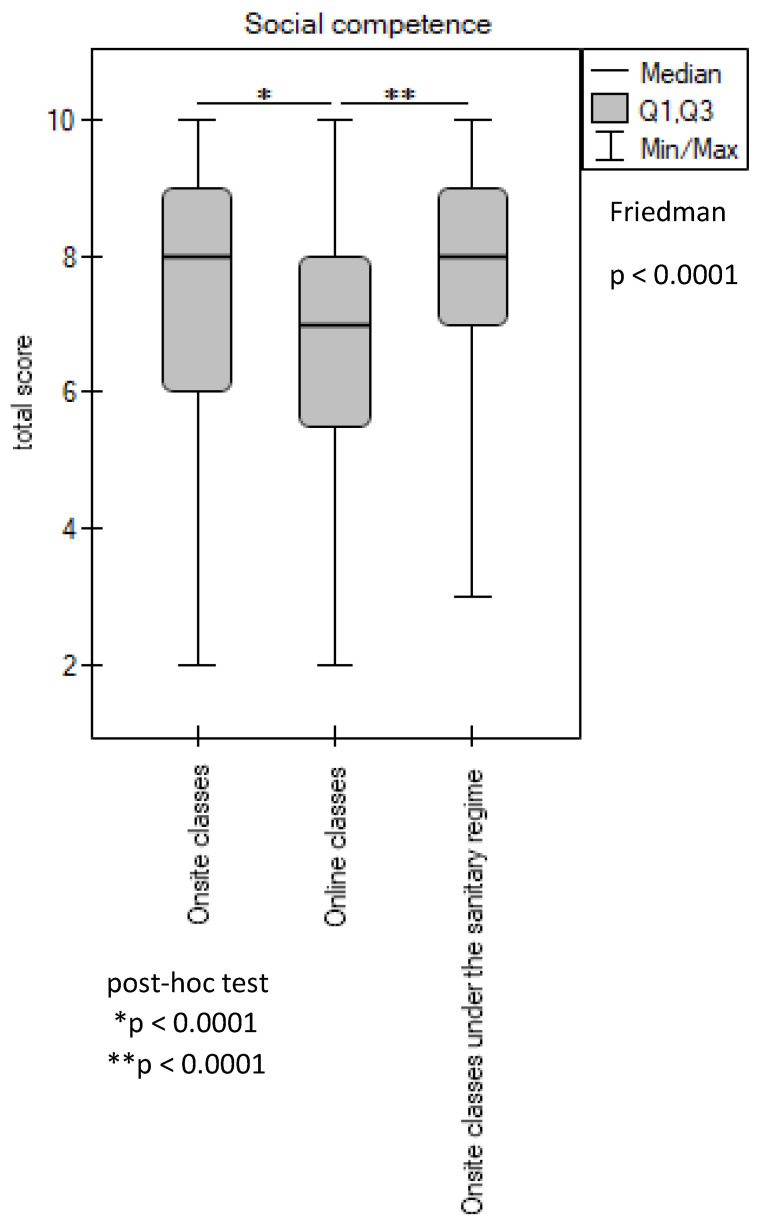
Evaluation of the learning outcomes in social competence. *, ** *p*-value is given for statistically significantly different groups. For groups with no significant differences (*p* ≥ 0.05), the *p*-values are not quoted.

**Figure 5 ijerph-20-02059-f005:**
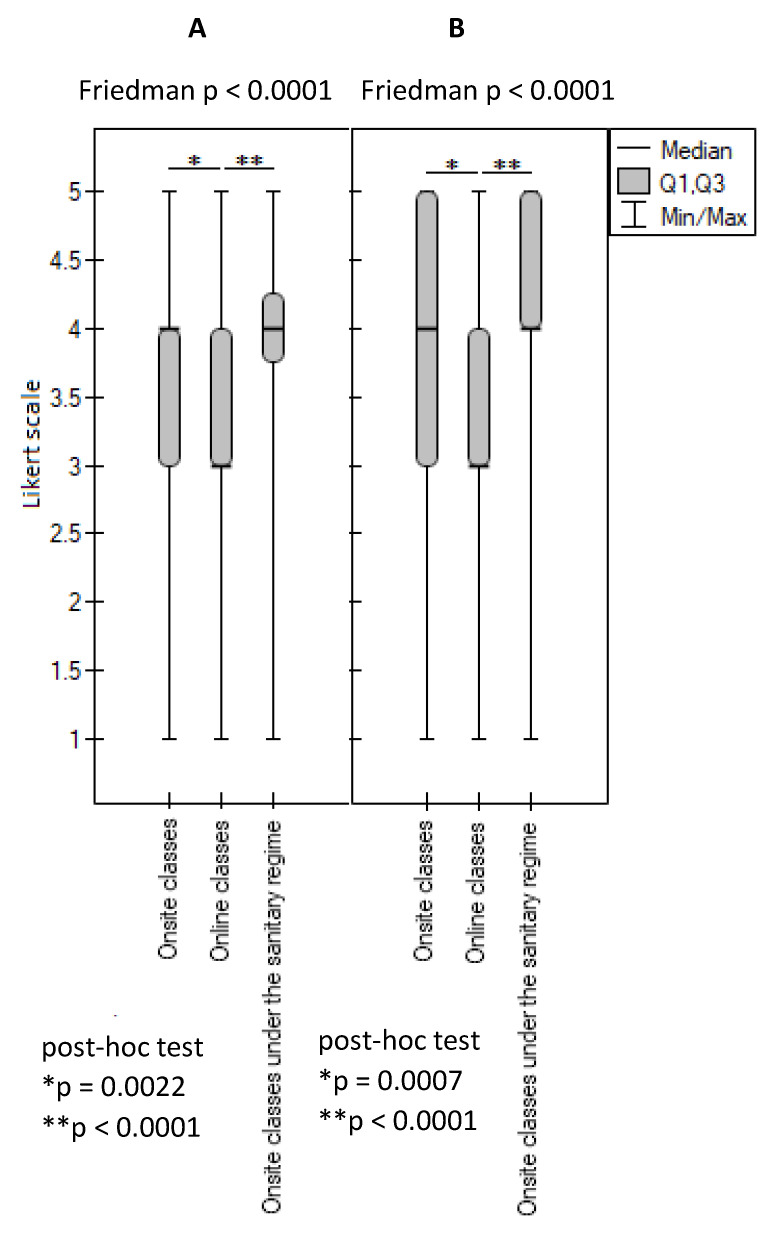
Evaluation of the learning outcomes achieved in the scope of: (**A**) communication with patients of the developmental age and their parents; (**B**) history taking from a patient and/or their family. *, ** *p*-value is given for statistically significantly different groups. For groups with no significant differences (*p* ≥ 0.05), the *p*-values are not quoted.

**Table 1 ijerph-20-02059-t001:** Percentage of students choosing a particular answer for a specific question (%).

Learning Outcomes	Survey Question	Score	Onsite Classes	Online Classes	Onsite Classes Under the Sanitary Regime
Knowledge	Did the clinical classes let you achieve the essential knowledge to perform the pediatric dentistry clinical procedures?	1	3.0	4.0	0.0
		2	4.0	13.1	6.1
		3	16.2	14.1	6.1
		4	49.5	39.4	59.6
		5	27.3	29.3	28.3
Skill	Did the clinical classes let you acquire the skills in clinical diagnosing of a patient in developmental age?	1	2.0	9.2	0.0
		2	5.1	14.3	5.1
		3	13.3	16.3	3.1
		4	41.8	30.6	52.0
		5	37.8	29.6	39.8
	Did the classes let you acquire the skills in applying appropriate methods and medications during and following dental procedures to relieve pain and anxiety in an adolescent patient?	1	4.2	9.5	1.1
		2	9.5	12.6	7.4
		3	14.7	17.9	9.5
		4	47.4	41.1	51.6
		5	24.2	18.9	30.5
	Did the clinical classes let you acquire the skills in prevention of oral diseases and conducting dental prophylaxis in a patient of developmental age?	1	2.1	6.3	0.0
		2	2.1	13.5	1.0
		3	12.5	8.3	6.3
		4	35.4	38.5	40.6
		5	47.9	33.3	52.1
	Did the classes let you acquire the skills in treating dental diseases in patients of developmental age?	1	4.1	14.3	2.0
		2	4.1	12.2	0.0
		3	11.2	15.3	10.2
		4	48.0	32.7	46.9
		5	32.7	25.5	40.8
	Did the classes let you acquire the skills in planning comprehensive treatment of the diseases of teeth, periodontal tissues and oral mucosa in children and adolescents?	1	3.1	10.2	2.0
		2	14.3	16.3	13.3
		3	24.5	22.4	20.4
		4	41.8	35.7	42.9
		5	16.3	15.3	21.4
	Did the classes let you acquire the skills in keeping records of an adolescent patient, writing letters of referral for them for specialized dental care, conducting general medical examinations or providing treatment?	1	5.2	13.4	3.1
		2	13.4	18.6	12.4
		3	38.1	34.0	28.9
		4	25.8	20.6	32.0
		5	17.5	13.4	23.7
Social competencies	Did the classes let you acquire the competences in communication with a patient of developmental age and their parents?	1	4.0	10.0	1.0
		2	5.0	14.0	5.0
		3	27.0	33.0	19.0
		4	40.0	29.0	50.0
		5	24.0	14.0	25.0
	Did the classes let you acquire the competences of history taking from a patient and/or their family?	1	3.0	14.1	1.0
		2	1.0	8.1	3.0
		3	27.3	30.3	16.2
		4	42.4	35.4	50.5
		5	26.3	12.1	29.3

## Data Availability

No new data were created or analyzed in this study.

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
