# Peer review of "Evaluation of Undergraduate Learning Experiences in Pediatric Dentistry during the COVID-19 Pandemic"

_ijerph, 2023, doi:10.3390/ijerph20032059_

Round 1
Reviewer 1 Report
The aim of the study was to analyze the students' opinions on the learning outcomes they achieved during clinical classes in pediatric dentistry. The topic is very interesting. Nevertheless, I suggest some improvements before its publication.
Abstract:
ICT is used but not yet defined.
It is defined at line 50-1 but shall be defined in the abstract. If published and searched through Pubmed or Scopus nobody will understand what ICT means.
Lines 143-145:
The authors wrote:
“Although the Friedman’s test showed statistically significant differences between three types of education (p=0.0162), a post-hoc test was not statistically significant (p-value >0.05). “
The authors should check better the statistical analysis: a post-hoc test should detect a significant difference outlined by the previous test.
Line 390:
The authors could also cite another study that confirms the importance of on-line learning during the pandemic.
The authors could add a sentence like the following.
“Furthermore, it has also been reported that almost 70% of dentists and hygienists attribute great importance to online courses”. Reference:
Paolone G, Mazzitelli C, Formiga S, Kaitsas F, Breschi L, Mazzoni A, Tete G, Polizzi E, Gherlone E, Cantatore G. 1 year impact of COVID-19 pandemic on Italian dental professionals: a cross-sectional survey. Minerva Dent Oral Sci. 2021 Dec 1. doi: 10.23736/S2724-6329.21.04632-5. Epub ahead of print. PMID: 34851068.
Please add the limitations of this study at the end of the discussion.
Please add further studies to be performed in the future on this topic.
Author Response
We would like to thank the reviewer for the careful and thorough reading of this manuscript and for invaluable comment. The following is our response (the reviewer’s comment is in italics):
The aim of the study was to analyze the students' opinions on the learning outcomes they achieved during clinical classes in pediatric dentistry. The topic is very interesting. Nevertheless, I suggest some improvements before its publication.
Response:
We appreciate the positive feedback from the reviewer. As suggested by the reviewer, we have carefully reviewed the manuscript.
Abstract:
ICT is used but not yet defined.
It is defined at line 50-1 but shall be defined in the abstract. If published and searched through Pubmed or Scopus nobody will understand what ICT means.
Response:
We fully agree with the reviewer's comment and apologize for this mistake. Thank you very much for pointing out this problem. In the revised version, we added the definition of ICT in the abstract.
Lines 143-145:
The authors wrote:
“Although the Friedman’s test showed statistically significant differences between three types of education (p=0.0162), a post-hoc test was not statistically significant (p-value >0.05). “
The authors should check better the statistical analysis: a post-hoc test should detect a significant difference outlined by the previous test.
Response:
We apologize for this mistake. We fully agree with the reviewer's comment. Thank you very much for pointing out this problem. Your suggestion is very valuable. We have carefully checked the statistical analysis and revised Fig.1, and this part of the Results section, which has been rewritten as:
Statistical analysis of the results revealed that the students rated their chances of obtaining the required knowledge higher during practical classes conducted onsite during the pandemic period compared to ICT classes (p = 0.0039). Moreover, in the students’ opinion, the face-to-face classes conducted before and during the pandemic enabled them to achieve learning outcomes in the scope of knowledge comparable extent (p > 0.05) (Fig.1).
Although the Friedman’s test showed statistically significant differences between three types of education (p=0.0162), a post-hoc test was not statistically significant (p-value >0.05) (Fig. 1).
Line 390:
The authors could also cite another study that confirms the importance of on-line learning during the pandemic.
The authors could add a sentence like the following.
“Furthermore, it has also been reported that almost 70% of dentists and hygienists attribute great importance to online courses”. Reference:
Paolone G, Mazzitelli C, Formiga S, Kaitsas F, Breschi L, Mazzoni A, Tete G, Polizzi E, Gherlone E, Cantatore G. 1 year impact of COVID-19 pandemic on Italian dental professionals: a cross-sectional survey. Minerva Dent Oral Sci. 2021 Dec 1. doi: 10.23736/S2724-6329.21.04632-5. Epub ahead of print. PMID: 34851068.
Response:
Thank you for your valuable comment. We have added the suggested reference.
Please add the limitations of this study at the end of the discussion. Please add further studies to be performed in the future on this topic.
Response:
Thank you very much for this comment. Your suggestion is very valuable. In the revised paper, we have added the following sentences:
The survey was attended by students who completed the entire course and passed the diploma exam in pediatric dentistry. After the study, all respondents no longer had classes in pediatric dentistry. In addition, before the study, all students were informed that the data collection is anonymous and that it is impossible to identify the person providing the answer. In this way, bias in answering was prevented. A limitation of the study is that it was conducted at only one dental school. The second limitation is the size of the research sample. The questionnaires were completed by 64% of the students invited to participate in the study. In this study, only students expressed opinions on the learning outcomes they achieved during clinical classes in pediatric dentistry during the pandemic period. These were the subjective opinions of students. The research will continue to determine teachers' views on students' learning outcomes.

Reviewer 2 Report
Comments on Kamienska et al:
The aim of this manuscript is to collect and analyze students’ opinions, regarding learning outcomes they achieved during clinical classes in pediatric dentistry.
This manuscript shows rich content, providing a deep insight for some works: the study is within the journal’s scope, and I found it to be well-written, providing sufficient information. Even if the manuscript provides an organic overview, with a densely organized structure and based on well-synthetized evidence, there are some suggestions necessary to make the article complete and fully readable. For these reasons, the manuscript requires major changes.
Please find below an enumerated list of comments on my review of the manuscript:
INTRODUCTION:
LINE 51: Coronavirus disease (COVID-19) is a global health threat, which spread through respiratory droplets and aerosol transmission. The causative agent of this disease is the severe acute respiratory syndrome coronavirus-2 (SARS-CoV-2), an enveloped positive single-stranded RNA virus, responsible of infection in pulmonary and extrapulmonary organs (see, for reference: Deshmukh, V.; Motwani, R.; Kumar, A.; Kumari, C.; Raza, K. Histopathological observations in COVID-19: A systematic review. J. Clin. Pathol. 2021, 74, 76–83).
LINE 56: In this perspective, dental schools were particularly affected by COVID-19 pandemic, due to the high risk of exposure of dental operators and dental students, during training practice. For these reasons, lectures and practical training were suspended and new methods of teaching were introduced (see, for reference: Varvara, G.; Bernardi, S.; Bianchi, S.; Sinjari, B.; Piattelli, M. Dental Education Challenges during the COVID-19 Pandemic Period in Italy: Undergraduate Student Feedback, Future Perspectives, and the Needs of Teaching Strategies for Professional Development. Healthcare 2021, 9, 454. https://doi.org/10.3390/healthcare9040454).
LINE 251: Please, reformulate this sentence by using the verb “provide”, instead of “equip”.
The main topic is interesting, and certainly of great clinical impact. As regards the originality and strengths of this manuscript, this is a significant contribute to the ongoing research on this topic, as it extends the research field on the impact of new learning experiences in pediatric dentistry, during the COVID-19 pandemic.
There is a specific and detailed explanation for the methods used in this study: this is particularly significant, since the manuscript relies on a multitude of recent evidence, to derive its conclusions.
The conclusion of this manuscript is perfectly in line with the main purpose of the paper: the authors have designed and conducted the study properly. As regards the conclusions, they are well written and present an adequate balance between the description of previous findings and the results presented by the authors.
Finally, this manuscript also shows a basic structure, properly divided and looks like very informative on this topic. Furthermore, figures and tables are complete, organized in an organic manner and easy to read.
In conclusion, this manuscript is densely presented and well organized, based on well-synthetized evidence. The authors were lucid in their style of writing, making it easy to read and understand the message, portrayed in the manuscript. Besides, the methodology design was appropriately implemented within the study. However, many of the topics are very concisely covered. This manuscript provided a comprehensive analysis of current knowledge in this field. Moreover, this research has futuristic importance and could be potential for future research. However, major concerns of this manuscript are with the introductive section and in the discussion: for these reasons, I have major comments for these sections, for improvement before acceptance for publication. The article is accurate and provides relevant information on the topic and I have some major points to make, that may help to improve the quality of the current manuscript and maximize its scientific impact. I would accept this manuscript if the comments are addressed properly.
Author Response
We sincerely thank the reviewer for constructive criticisms and comments, which were of great help in revising the manuscript. We have revised our manuscript and rewritten the introductions and discussion sections. Please, find below our responses (the reviewer’s comments are in italics).
The aim of this manuscript is to collect and analyze students’ opinions, regarding learning outcomes they achieved during clinical classes in pediatric dentistry.
This manuscript shows rich content, providing a deep insight for some works: the study is within the journal’s scope, and I found it to be well-written, providing sufficient information. Even if the manuscript provides an organic overview, with a densely organized structure and based on well-synthetized evidence, there are some suggestions necessary to make the article complete and fully readable. For these reasons, the manuscript requires major changes.
Please find below an enumerated list of comments on my review of the manuscript:
INTRODUCTION:
LINE 51: Coronavirus disease (COVID-19) is a global health threat, which spread through respiratory droplets and aerosol transmission. The causative agent of this disease is the severe acute respiratory syndrome coronavirus-2 (SARS-CoV-2), an enveloped positive single-stranded RNA virus, responsible of infection in pulmonary and extrapulmonary organs (see, for reference: Deshmukh, V.; Motwani, R.; Kumar, A.; Kumari, C.; Raza, K. Histopathological observations in COVID-19: A systematic review. J. Clin. Pathol. 2021, 74, 76–83).
LINE 56: In this perspective, dental schools were particularly affected by COVID-19 pandemic, due to the high risk of exposure of dental operators and dental students, during training practice. For these reasons, lectures and practical training were suspended and new methods of teaching were introduced (see, for reference: Varvara, G.; Bernardi, S.; Bianchi, S.; Sinjari, B.; Piattelli, M. Dental Education Challenges during the COVID-19 Pandemic Period in Italy: Undergraduate Student Feedback, Future Perspectives, and the Needs of Teaching Strategies for Professional Development. Healthcare 2021, 9, 454. https://doi.org/10.3390/healthcare9040454).
Response:
Thank you for your valuable comment. We have added both references.
LINE 251: Please, reformulate this sentence by using the verb “provide”, instead of “equip”.
Response:
Thank you very much for pointing out this problem. We have revised the sentence.
The main topic is interesting, and certainly of great clinical impact. As regards the originality and strengths of this manuscript, this is a significant contribute to the ongoing research on this topic, as it extends the research field on the impact of new learning experiences in pediatric dentistry, during the COVID-19 pandemic.
There is a specific and detailed explanation for the methods used in this study: this is particularly significant, since the manuscript relies on a multitude of recent evidence, to derive its conclusions.
The conclusion of this manuscript is perfectly in line with the main purpose of the paper: the authors have designed and conducted the study properly. As regards the conclusions, they are well written and present an adequate balance between the description of previous findings and the results presented by the authors.
Finally, this manuscript also shows a basic structure, properly divided and looks like very informative on this topic. Furthermore, figures and tables are complete, organized in an organic manner and easy to read.
In conclusion, this manuscript is densely presented and well organized, based on well-synthetized evidence. The authors were lucid in their style of writing, making it easy to read and understand the message, portrayed in the manuscript. Besides, the methodology design was appropriately implemented within the study. However, many of the topics are very concisely covered. This manuscript provided a comprehensive analysis of current knowledge in this field. Moreover, this research has futuristic importance and could be potential for future research. However, major concerns of this manuscript are with the introductive section and in the discussion: for these reasons, I have major comments for these sections, for improvement before acceptance for publication. The article is accurate and provides relevant information on the topic and I have some major points to make, that may help to improve the quality of the current manuscript and maximize its scientific impact. I would accept this manuscript if the comments are addressed properly.
Response:
Thank you very much for this comment. We have carefully revised the manuscript. We hope we have improved the manuscript by adding some sentences and clarifying the text.

Reviewer 3 Report
Introduction
The introduction was compact, informative and easy to read.
Materials and Methods
1. No information is given on the development of the questionnaire. Was it adapted from past studies (If yes, please cite the study) or was it developed by the authors? If it was developed by the authors, were the validity and reliability checked?
2. After the questionnaire was developed, was a pilot test conducted? Please provide reasons if not done.
3. Since semester VIII was delivered entirely via the online method, would it be biased to ask the students to assess if they have achieved certain practical skills and social competence under this setting?
Results
1. Typo in table 1 (Skill)
Discussion
1. Discuss the limitations of the study, taking into account sources of potential bias or imprecision.
Author Response
We would like to thank the reviewer for the careful and thorough reading of this manuscript and for the invaluable comments and constructive suggestions. Please find our point-by-point responses (the reviewer’s comments are in italics).
The introduction was compact, informative and easy to read.
Materials and Methods
- No information is given on the development of the questionnaire. Was it adapted from past studies (If yes, please cite the study) or was it developed by the authors? If it was developed by the authors, were the validity and reliability checked?
- After the questionnaire was developed, was a pilot test conducted? Please provide reasons if not done.
Response:
Thank you very much for this comment. We agree with the reviewer and we have rewritten the materials and methods section, and by adding the sentence to the Material and Methods section of the manuscript.
As all questions used in this study were developed by researchers specifically for this project and the recent outbreak of COVID-19, the questionnaire was not validated or piloted before this study.
- Since semester VIII was delivered entirely via the online method, would it be biased to ask the students to assess if they have achieved certain practical skills and social competence under this setting?
Semester VIII was the beginning of the COVID-19 pandemic in Europe. It was a very difficult time, also for dental education. Like numerous medical universities, Poznan Medical University decided to continue its education by implementing new online teaching techniques. Of course, it is not the same as contact with real patients, and this part of education cannot be replaced by ICT. The dental students in our University had clinical classes online as they had in the schedule. We have virtual meetings. Mostly we had presentations of the cases (also prepared by students), but we also had live broadcasts from the dental office of our dental school, where there was a teacher with the patient.
Results
Typo in table 1 (Skill)
Response:
We fully agree with reviewers comment and apologize for this mistakes
Discussion
Discuss the limitations of the study, taking into account sources of potential bias or imprecision.
Response:
Thank you very much for this comment. Your suggestion is very valuable. In the revised paper we have added the following sentences:
The survey was attended by students who completed the entire course and passed the diploma exam in pediatric dentistry. After the study, all respondents no longer had classes in pediatric dentistry. In addition, before the study, all students were informed that the data collection is anonymous and that it is impossible to identify the person providing the answer. In this way, bias in answering was prevented. A limitation of the study is that it was conducted at only one dental school. The second limitation is the size of the research sample. The questionnaires were completed by 64% of the students invited to participate in the study. In this study, only students expressed opinions on the learning outcomes they achieved during clinical classes in pediatric dentistry during the pandemic period. These were the subjective opinions of students.

Round 2
Reviewer 2 Report
Authors complied to the suggestions
Manuscript can be now accepted